# A Regulatory Loop Involving miR-200c and NF-κB Modulates Mortalin Expression and Increases Cisplatin Sensitivity in an Ovarian Cancer Cell Line Model

**DOI:** 10.3390/ijms232315300

**Published:** 2022-12-04

**Authors:** Xin Huang, Yichen Yan, Ailing Gui, Shun Zhu, Shi Qiu, Feng Chen, Wen Liu, Ji Zuo, Ling Yang

**Affiliations:** Department of Cellular and Genetic Medicine, School of Basic Medical Sciences, Fudan University, Shanghai 200032, China

**Keywords:** ovarian cancer, miR-200b/c, mortalin, NF-κB, drug resistance

## Abstract

Ovarian cancer is currently the most lethal gynecological cancer. At present, primary debulking surgery combined with platinum-based chemotherapy is the standard treatment strategy for ovarian cancer. Although cisplatin-based chemotherapy has greatly improved the prognosis of patients, the subsequent primary or acquired drug resistance of cancer cells has become an obstacle to a favorable prognosis. Mortalin is a chaperone that plays an important role in multiple cellular and biological processes. Our previous studies have found that mortalin is associated with the proliferation and migration of ovarian cancer cells and their resistance to cisplatin-based chemotherapy. In this study, microRNA (miR)-200b/c downregulated mortalin expression and inhibited the proliferation and migration of the paired cisplatin-sensitive (A2780S) and cisplatin-resistant (A2780CP) epithelial ovarian cancer cell lines. Moreover, miR-200c increased the sensitivity of ovarian cancer cells to cisplatin treatment by regulating mortalin levels. Nuclear factor (NF)-κB directly regulated mortalin and miR-200b/c expression levels, while NF-κB and miR-200b/c jointly regulated the expression of mortalin. The combination of cisplatin and miR-200c significantly enhanced the therapeutic effects on ovarian cancer in vivo, suggesting that miR-200c may serve as a potential therapeutic agent for ovarian cancer.

## 1. Introduction

Ovarian cancer is one of the most common gynecological cancers, with incidence and mortality rates of 3.4% and 4.4%, respectively, as of 2018 [1]. As there are, typically, few clinical symptoms in the early stages, many patients with ovarian cancer do not receive medical intervention until the disease has progressed to an advanced stage. More than 70% of patients with ovarian cancer are at stage III or IV, whereas only 15% are at stage I [2,3]. Primary cytoreductive surgery followed by platinum adjuvant chemotherapy is the standard treatment strategy for ovarian cancer at present. However, primary or acquired cisplatin resistance in ovarian cancer leads to poor prognosis [4,5]. Therefore, determining the molecular mechanisms of drug resistance and validating diagnostic and prognostic biomarkers will facilitate the development of better therapeutic strategies for ovarian cancer.

Mortalin is a molecular chaperone belonging to the heat shock protein 70 family (HSP70), also known as mitochondrial heat shock protein 70 (mtHSP70), peptide-binding protein 74 (PBP74), and glucose-regulated protein 75 (GRP75). Mortalin is highly expressed in various cancer types, including thyroid cancer, gastric cancer, and hepatobiliary carcinoma. An increased expression of mortalin often leads to a worse clinical prognosis [6,7,8]. Significantly, our previous study demonstrated that mortalin expression correlates with the clinical stage of patients with ovarian cancer [9]. Thus, inhibiting the expression of mortalin leads to decreased proliferation and migration and increased sensitivity to cisplatin in ovarian cancer cells [9,10]; however, the upstream regulation of mortalin is not yet fully understood.

MicroRNAs (miRNAs) are small non-coding RNAs composed of approximately 22 nucleotides that regulate gene expression at the post-transcriptional level [11]. miRNAs bind to the mRNAs of target genes via complete or incomplete complementarity, forming a miRNA-induced silencing complex which promotes the degradation of mRNA and represses protein translation [12,13,14]. miRNAs are abnormally expressed in various types of cancer. Some miRNAs promote cancer progression by targeting tumor suppressor genes, whereas others target oncogenes and inhibit the proliferation of cancer cells. The miR-200 family comprises five members (collectively known as miR-200s): miR-200a, miR-200b, miR-200c, miR-141, and miR-429. miR-200s exert anticancer effects against various cancer types. miR-200s hamper cancer progression by inhibiting epithelial-mesenchymal transition and angiogenesis [15,16,17,18]. miR-200c, a member of the miR-200 family, inhibits intratumoral angiogenesis in ovarian cancer by downregulating the expression of interleukin (IL)-8 and C-X-C motif chemokine ligand 1 (CXCL1) [19].

Nuclear factor (NF)-κB is a transcription factor that plays a crucial role in cancer progression. Our previous study showed that NF-κB p65 promotes the proliferation and migration of ovarian cancer cells by binding to the *mortalin* promoter region [20]. Accumulating evidence suggests that multiple transcription factors and miRNAs regulate the expression of most genes. Transcription factors and miRNAs co-regulate gene expression, forming a regulatory network that modulates various biological processes. In addition, there is a mutual regulatory relationship between transcription factors and miRNAs [21]. Studies have reported a mutual regulatory relationship between NF-κB and miR-200s. NF-κB binds to the promoter region of zinc finger E-box binding homeobox 1/2 and promotes its expression, thereby indirectly promoting the expression of miR-200-b and -c in breast cancer [22]. In contrast, miR-200b/c inhibit the activation of the NF-κB signaling pathway by targeting MyD88 [23]. However, the relationship between miR-200b/c and the NF-κB signaling pathway in ovarian cancer remains poorly understood.

In the current study, we found that miR-200c directly targeted mortalin, thereby suppressing the proliferation and migration of ovarian cancer cells and decreasing their drug resistance. Moreover, NF-κB p65 promoted miR-200b/c expression by binding to the promoter of *miR-200b/c* in ovarian cancer cells, consequently forming a regulatory loop.

## 2. Results

### 2.1. Mortalin Is a Direct Target of miR-200b/c

To identify the miRNAs targeting and regulating mortalin expression, four miRNA databases (TargetScan, miRanda, miRDB, and MicroCosm) were analyzed. Only 3 miRNAs (miR-200b/200c/429) of 807 were predicted across all four databases (Figure 1A). miR-4499 was predicted to regulate mortalin expression in the TargetScan and miRanda data base and was also tested for regulatory effects. Further experiments showed that the upregulation of miR-429 and miR-4499 did not significantly inhibit mortalin expression (Appendix A); therefore, miR-200b and miR-200c were chosen for the follow-up study. Sequence analysis showed that *mortalin* contains a possible target site for miR-200b/c in the 3′-Untranslated Regions (UTR) region. Luciferase reporter plasmids containing the *mortalin* 3′-UTR sequence (*mortalin* 3′-UTR wild-type group) and the putative binding site-mutated *mortalin* 3′-UTR sequence (*mortalin* 3′-UTR mutation group) were constructed (Figure 1B). Due to high transfection efficiency and strong protein expression levels, 293T cells were chosen to verify the direct interaction between miR-200b/c and mortalin [24]. The results of the dual-luciferase reporter assay showed that the fluorescence intensity was significantly decreased after transfection with the miR-200b/c mimic in the *mortalin* 3′-UTR wild-type group, but not the *mortalin* 3′-UTR mutation group (Figure 1C). To determine whether miR-200b/c could regulate the expression of mortalin in ovarian cancer cells, cisplatin-sensitive A2780S and cisplatin-resistant A2780CP ovarian cancer cell lines were chosen for further analysis. Treatment with the miR-200b/c mimic significantly reduced the mRNA and protein expression of *mortalin* (Figure 1D,E). Immunofluorescence assays also showed that transfection with miR-200b/c significantly decreased the protein expression of mortalin (Figure 1F,G). Though miR-200b and miR-200c have similar inhibition of mortalin expression in A2780S cells, miR-200c has a stronger inhibition of mortalin expression than miR-200b in A2780CP cells.

### 2.2. miR-200b/c Decrease the Proliferation and Migration of Ovarian Cancer Cells Via Mortalin

To determine the effect of miR-200b/c on ovarian cancer cells, cell proliferation and migration assays were performed. The cell counting kit (CCK)-8 assay showed that transfection with the miR-200b/c mimic significantly decreased the proliferation of both A2780S and A2780CP cells (Figure 2A). Simultaneously, mortalin overexpression rescued ovarian cancer cell proliferation (Figure 2A). The wound healing assay (Figure 2B) showed that transfection with miR-200b/c mimic significantly reduced the migration of ovarian cancer cells, which was rescued by mortalin overexpression. The transwell assay also confirmed that the miR-200b/c mimic significantly reduced the migration of both A2780S and A2780CP cells, which was rescued by mortalin overexpression (Figure 2C). In contrast, the inhibition of miR-200b/c with a miR-200b/c inhibitor resulted in the increased proliferation and migration of ovarian cancer cells (Appendix A). Western blotting was used to determine the effect of miR-200b/c on the expression and activation of Akt and extracellular signal-regulated protein kinase (Erk). Results (Figure 2D) showed that transfection with the miR-200b/c mimic significantly reduced the phosphorylation of Erk. Moreover, both miR-200b and miR-200c decreased the total Akt and phosphorylated Akt levels in A2780S and A2780CP cells. 

### 2.3. miR-200c Increases the Sensitivity of Ovarian Cancer Cells to Cisplatin

Cisplatin is one of the most important drugs in ovarian cancer treatment. A2780CP is a cisplatin-resistant ovarian cancer cell line, whereas the A2780S cell line is sensitive to cisplatin treatment. Similar to the results of a previous study, the half-maximal inhibitory concentration of cisplatin in A2780CP cells was significantly higher than that in A2780S cells (Appendix A). The mRNA and protein levels of *mortalin* in A2780CP cells were significantly higher than those in A2780S cells (Appendix A). However, the transcript levels of *miR-200b/c* in A2780CP cells were significantly lower than those in A2780S cells (Figure 3A), and the opposite was found for *mortalin* expression levels (Appendix A). To determine the effect of miR-200b/c on cisplatin resistance in ovarian cancer cells, cell viability was determined after transfection with miR-200b/c mimic with or without cisplatin for 24 h. miR-200c transfection led to decreased cell viability (Figure 3B), an increased ratio of apoptotic cells (Figure 3C), and the increased expression of cleaved caspase 3 and poly (ADP ribose) polymerase (PARP)-1 (Figure 3D,E). Previous studies have reported that mortalin promotes cisplatin resistance by inhibiting the entry of p53 into the nucleus and promoting its degradation [9]. Consistently, miR-200c transfection increased the overall expression of p53 and its translocation into the nucleus after cisplatin treatment in this study (Figure 3F). However, miR-200b transfection did not affect the apoptosis of ovarian cancer cells and the expression and location of p53 after cisplatin treatment (Figure 3C,F). To determine whether miR-200c overcame cisplatin resistance in A2780CP cells via mortalin, a mortalin overexpression plasmid was transfected into A2780CP cells (Figure 3G). Compared with the control group, A2780CP cells transfected with the miR-200c mimic were found to be significantly more sensitive to cisplatin, which could be rescued by co-transfection with the mortalin overexpression plasmid (Figure 3H). This, suggests that miR-200c increases sensitivity to cisplatin treatment via mortalin.

### 2.4. miR-200c Suppresses the Chemoresistance of Ovarian Cancer Cells In Vivo

To demonstrate the role of miR-200b/c in the chemoresistance of ovarian cancer cells in vivo, ovarian cancer cells were subcutaneously injected into female nude mice. AgomiR is a chemically modified miRNA agonist with high stability and activity. AgomiR has been widely used in vivo to increase miRNA expression [25]. Mice were injected intratumorally with agomiR-200b/c or NC to induce the local expression of miR-200b/c in nude mice (Figure 4A). The tumor growth curve showed that the subcutaneous tumors treated with agomiR-200c showed significantly decreased tumor growth and reduced tumor volume and mass (Figure 4B–D). However, agomiR-200b treatment did not significantly influence the growth of subcutaneous tumors, although a decreasing trend was observed. Importantly, treatment with agomiR-200b/c led to a significant reduction in mortalin mRNA and protein levels in isolated ovarian cancer tissues (Figure 4E,F). It should be mentioned that the tumor size of mice in the no-treatment group exceeded 2000 mm^3^ at day 19, so these mice were sacrificed for ethical reasons.

### 2.5. miR-200c Expression Levels Are Associated with Patient Prognosis

Ovarian cancer data with clinically annotated samples from the Cancer Genome Atlas (TCGA) were analyzed to evaluate the effect of miR-200b/c levels on patient prognosis. The transcript levels of *miR-200c* were lower in patients at stage IV than those at stage II. Additionally, the transcription levels of *miR-200c* trended to be lower in patients at stage III than those at stage II, although there was no statistical significance (Figure 5A). *miR-200c* levels were positively correlated with the overall survival of patients with ovarian cancer (Figure 5B). Patients with the *miR-200c* gene mutation had lower overall survival than those with the wild-type *miR-200c* genotype (Figure 5C). However, *miR-200b* levels were not significantly correlated with the prognosis or disease stage of patients with ovarian cancer. In addition, *miR-200b* and *miR-200c* levels were negatively correlated with *mortalin* expression in patients with ovarian cancer, but only *miR-200c* was statistically significant (Figure 5D). These results demonstrate a significant correlation between *miR-200c* levels and the prognosis of patients with ovarian cancer, suggesting that miR-200c may serve as a candidate prognostic marker for ovarian cancer.

### 2.6. NF-κB and miR-200b/c Co-Regulate Mortalin Expression

Our previous study revealed that mortalin expression is regulated by the transcription factor NF-κB p65 [20]. However, the mutual regulatory relationship between NF-κB and miR-200b/c remained unclear. Here, we found that transfection with miR-200b/c did not affect NF-κB p65 mRNA levels or translocation into the nucleus (Figure 6A,B). In contrast, *miR-200b/c* levels significantly increased upon NF-κB overexpression (Figure 6C). Considering that there is a putative NF-κB binding site in the *miR-200b/c* promoter region (Figure 6D), the chromatin Immunoprecipitation (ChIP) assay was performed to detect direct binding. The ChIP assay showed that the protein–DNA complex immunoprecipitated with the NF-κB p65 antibody, leading to specific *miR-200b/c* PCR products (Figure 6E). To demonstrate that NF-κB p65 and miR-200b/c coregulate mortalin expression, either the miR-200b/c mimic or NC was transfected into A2780CP cells stably overexpressing NF-κB p65. Mortalin levels were increased in the NF-κB-overexpressing cell line, which was decreased by miR-200b/c expression (Figure 6F). Immunofluorescence assay showed that NF-κB p65 overexpression reduced p53 nuclear translocation, while co-transfection of miR-200b/c mimic significantly increased translocation of p53 into the nucleus (Figure 6G). The wound healing assay showed NF-κB p65 overexpression increased the migration of ovarian cancer cells, which was reduced by miR-200b/c mimic co-miR-200b/c. (Figure 6H). Overall, above results suggest that NF-κB p65 and miR-200b/c co-regulate mortalin expression and the following p53 nuclear translocation and ovarian cancer cells migration.

## 3. Discussion

Mortalin, which plays an important role in the pathophysiological processes of cells, is involved in the occurrence of multiple diseases, including degenerative diseases and cancer. Ovarian cancer has the highest mortality rate among all gynecological malignancies. Previous studies have shown that mortalin is a potential therapeutic target for ovarian cancer. In this study, we uncovered a novel regulatory mechanism of mortalin mediated by miR-200b/c and NF-κB, and determined its effects on ovarian cancer. miR-200c treatment decreased mortalin expression and suppressed ovarian cancer malignancy (Figure 7). Through bioinformatics analysis, we found that miR-200b/c might regulate the expression of mortalin, and a luciferase reporter gene assay further confirmed that miR-200b/c directly binds to the 3′-UTR region of *mortalin*. Overexpression of miR-200b/c decreased the mRNA and protein levels of *mortalin* in ovarian cancer cells. A previous study also reported a conserved site of the miR-200 family in the 3′-UTR region of *mortalin*, which was predicted using TargetScan software [26]. Inhibition of miR-200b/c and miR-217 expression leads to increased mRNA levels of *mortalin* without affecting its protein levels in human chronic myeloid leukemia cells [27].

Cisplatin is a first-line chemotherapeutic drug widely used to treat high-grade malignant cancers [28]. Previous studies have shown that cisplatin is effective at the time of initial drug use. However, drug resistance has been observed in patients with relapse, which significantly affects the long-term efficacy of cisplatin [29]. A combination of cisplatin and other drugs can reduce drug tolerance and toxicity. By analyzing TCGA database, we found that patients with high clinical-stage ovarian cancer had decreased levels of *miR-200c* expression. miR-200c increased the sensitivity of ovarian cancer cells to cisplatin by regulating mortalin expression. To further explore the medicinal value of miR-200b/c and the effect of miR-200c on cisplatin resistance in vivo, we used a subcutaneous tumorigenesis nude mouse model. The combined application of cisplatin and agomiR-200c significantly enhanced the therapeutic effects of cisplatin. We also found that agomiR-200c treatment inhibited mortalin expression in ovarian cancer tissues. In the present and previous studies, no obvious side effects were observed after the intratumoral injection of miR-200c, indicating its safety, which suggests the potential use of miR-200c as a potential therapeutic target for ovarian cancer.

miR-200b and miR-200c belong to the miR-200 family and have the same seed-binding sequence [30]. Several studies have reported that miR-200b/c can reduce cell proliferation and slow tumor growth [31,32,33]. In this study, both miR-200b and miR-200c reduced the proliferation and migration of ovarian cancer cells, but only miR-200c increased the sensitivity of ovarian cancer to cisplatin. A previous study reported that miR-200c is the most efficient miRNA targeting mortalin using Microcosom Target [26]. We found that miR-200b and miR-200c have similar inhibition of mortalin expression in A2780S cells, however, miR-200c has a stronger inhibition of mortalin expression than miR-200b in A2780CP cells. Meanwhile, miR-200c rather than miR-200b increased p53 expression and nuclear translocation after cisplatin treatment. Different independent studies have found that mortalin interacts with p53 and sequesters it in the cytoplasm [34,35,36]. Additionally, our previous study also confirms that inhibition of mortalin increases the translocation of p53 to the nucleus, which partially enhances the sensitivity of ovarian cancer cells to cisplatin [9]. Thus, miR-200c may increase the sensitivity of ovarian cancer cells to cisplatin partly by increasing the nuclear translocation of p53 via mortalin. Furthermore, in ovarian cancer, miR-200c has more tumor suppressor targets than miR-200b. miR-200c, but not miR-200b, can target oncogenes, such as *HuR* [37], *TUBB3* [38], *CXCL1*, and *IL-8* [19]. miR-200c sensitizes Olaparib-resistant ovarian cancer cells by targeting Neuropilin 1 [39], which has been shown to promote unlimited growth through the evasion of contact inhibition [40]. Therefore, besides the mortalin-p53 axis, miR-200c may also affect cisplatin resistance in ovarian cancer through other proteins and pathways. At the same time, the *miR-200b* levels did not correlate with the clinical stage or prognosis of patients, suggesting that miR-200c is a more suitable drug target and molecular marker for patients with ovarian cancer. However, further studies are required to understand the different effects of miR-200b and miR-200c on drug resistance of ovarian cancer cells.

Our previous study showed that NF-κB p65 promotes the expression of mortalin by binding to its promoter region [20]. However, mortalin overexpression only partly reverses the decreased proliferation and migration ability of ovarian cancer cells induced by NF-κB p65 downregulation. Thus, other factors may also regulate mortalin in ovarian cancer cells. Gene expression is often regulated by multiple trans-acting factors, including transcription factors and microRNAs [41,42]. While co-regulating target genes, transcription factors, and microRNAs also have mutual regulatory relationships [43]. Therefore, transcription factors, microRNAs and their target genes form a basic regulatory unit, also known as network motifs, which participate in the complex gene transcription regulatory network of cells [44]. A previous study reported a feed-forward loop between p53 miRNA and E2F in breast cancer [45]. p53 inhibits the activation of transcription factor E2F and regulates the expression of various miRNAs in an E2F-dependent manner, and there is also a two-way regulatory relationship between miRNAs and E2F, which jointly regulates the downstream genes that inhibit cancer proliferation, forming a regulatory loop. Another study documented a feed-forward loop among miR-15/16, transcription factor E2F1, and cyclin E. Cyclin E is regulated by both miR-15/16 and E2F1, and E2F1 further promotes the expression of miR-15/16. Therefore, miR-15/16 inhibits E2F1-induced cell proliferation by targeting cyclin E [46]. A negative feedback regulatory loop between miR-200b and NF-κB has been reported. NF-κB binds to the promoter region of *miR-200b* increasing its expression. In contrast, miR-200b targets the inhibitor of nuclear factor kappa B kinase subunit beta and indirectly inhibits the activation of NF-κB p65 [47]. In this study, we explored the regulatory relationship between NF-κB p65 and miR-200b/c in ovarian cancer. NF-κB p65 increased miR-200b/c expression by binding to the *miR-200b/c* promoter in ovarian cancer cells. However, miR-200b/c did not affect NF-κB p65 expression. NF-κB p65 and miR-200b/c co-regulate mortalin expression, forming a regulatory loop that influences p53 translocation to nuclear and the migration of ovarian cancer cells. Interesting, the regulatory loop in our study belongs to one type of Feed Forward Loop (FFL), namely FFL (TF→miR) [21], because transcription factor NF-κB p65 regulates the expression of mortalin and miR200b/c, and miR200b/c, in turn, inhibits mortalin expression (Figure 7). Therefore, we reasonably speculate NF-κB p65 plays a predominant role in regulating mortalin in A2780CP, while the miR-200b/c is a negative feedback to modulate this process. Additionally, the expression of miR-200b/c may be affected by other regulatory factors. Our study only preliminarily indicates a regulatory loop, in which there is an interaction among NF-κB p65, miR-200b/c, and mortalin; however, their impact on the occurrence and development of ovarian cancer requires further elucidation.

However, there are still some limitations of this study. Ovarian cancer develops due to a complex interplay between hereditary and environmental factors. Although often described as one disease, ovarian cancer is actually a group of distinct tumor types [48]. In ovarian cancers, tumor heterogeneity appears to be very high across subtypes and within a single tumor, representing a major cause of treatment failure [49]. However, our study explores a specific mechanism of gene regulation in only one cell line model. Therefore, whether this mechanism is universal in ovarian cancer needs to be further studied.

In summary, our study provides novel insight into the previously unrecognized mechanism of mortalin regulation that alters the tumor malignancy in ovarian cancer. Collectively, our data suggest that miR-200c is a potential therapeutic target for ovarian cancer treatment.

## 4. Materials and Methods

### 4.1. Cell Culture and Transfection

The human ovarian cancer cell line A2780S (93112519) and the cisplatin resistant cell line A2780CP (93112517) were obtained from the European Collection of Authenticated Cell Cultures (ECACC). The human embryonic kidney 293T cell line was obtained from American type culture collection (ATCC) (CRL-3216). All the cell lines were grown at 37 °C in a humidified atmosphere of 5% CO_2_. The cells were cultured in DMEM medium (meilunbio, Dalian, China) supplemented with 10% FBS (yeasen, Shanghai, China). miR-200b/c, miR-429, miR-4499 mimics or the negative control (NC) (RiboBio, Guangzhou, China) were transfected into the cells at a final concentration of 50 nM using the Hieff Trans^TM^ liposomal transfection reagent (yeasen, Shanghai, China) according to the manufacturer’s instructions.

### 4.2. Dual Luciferase Reporter Assay and Mortalin 3′-UTR Site Mutagenesis

Mortalin 3′-UTR sequences were cloned into the pmiRGLO luciferase reporter (Youbio, Changsha, China). Then, 293T cells were co-transfected with pmiRGLO reporter and NC or miR-200b/c mimic (50 nM). After 24 h of cell culture, the dual-luciferase reporter gene assay (Beyotime, Shanghai, China) was performed, and the firefly luciferase and Renilla luciferase signals were measured using a Multiskan MK3 microplate reader (Thermo Fisher Scientific, Waltham, MA, USA), according to the manufacturer’s guidelines. Firefly luciferase activity was normalized to Renilla luciferase activity. The ratios were normalized to those of the NC.

Mutant mortalin 3′-UTR was constructed for the predicted binding site mentioned in Figure 1B using the Mut Express II Fast Mutagenesis Kit (Vazyme, Nanjing, China) with forward and reverse primer (5′-TGATAGACGCCAGCATGTGCAAATCTTGTTTGAAGG-3′) and a reverse primer (5′-GCACATGCTGGCGTCTATCATTATTTCATTATAACAGCCCTTCCAAATCTTGA-3′). The mutation was confirmed via DNA sequencing before using the mutant mortalin 3′-UTR constructs for the luciferase assay.

### 4.3. RNA Isolation and Real-Time PCR Assay

Total RNA was isolated from the cells or tumors using Trizol reagent (Vazyme, Nanjing, China) according to the manufacturer’s protocol. cDNA was synthesized using a HiScript III 1st Strand cDNA Synthesis Kit (Vazyme, Nanjing, China). qPCR was performed using the Hieff^®^ qPCR SYBR Green Master Mix (High Rox Plus) (yeasen, Shanghai, China) on a LightCycler Nano real-time fluorescence quantification PCR system (Roche, Indianapolis, IN, USA). All primer sequences used for qPCR are listed in the Appendix A. *miR-200b* (MQPS0000781-1-100), *miR-200c* (MQPS0000783-1-100), and *U6* (MQPS0000002-1-100) primers for qPCR were purchased from RiboBio company. U6 small nuclear RNA was used as an internal control to normalize miR-200b/c expression level. GAPDH was used as an internal control to normalize the expression levels of mortalin and NF-κB p65.

### 4.4. Western Blot Analysis

The total protein was subjected to sodium dodecyl sulfate-polyacrylamide gel electrophoresis (meilunbio, Dalian, China) and transferred to polyvinylidene fluoride membranes (Millipore, Burlington, MA, USA). Subsequently, the membranes were blocked with 5% skimmed milk before incubation with a 1:3000 dilution of primary antibodies at 4 °C overnight, followed by incubation with a 1:5000 dilution of secondary antibodies (Proteintech, Chicago, IL, USA) for 2 h at room temperature. Finally, the chemiluminescence signal was detected using a super sensitive chemiluminescence reagent (meilunbio, Dalian, China) and the signal intensity of the target protein was determined using a Gel Doc XR System (Bio-Rad, Hercules, CA, USA). The results were analyzed using Image J software (National Institutes of Health, Bethesda, ML, USA). The following primary antibodies were used: anti-MORTALIN (Cell Signaling Technology, Boston, MA, USA, 3593S), anti-GAPDH (Proteintech, Chicago, IL, USA, 6004-1-Ig), anti-PARP1 (Proteintech, Chicago, IL, USA, 13371-1-AP), anti-CASPASE3 (Proteintech, Chicago, IL, USA, 19677-1-AP), anti-p53 (Proteintech, Chicago, IL, USA, 60283-2-Ig), anti-p65 (Proteintech, Chicago, IL, USA, 10745-1-AP), anti-Erk1/2 (Bioss, Beijing, China, bsm-53359R), anti-Phospho-Erk1/2 (Proteintech, Chicago, IL, USA, 28733-1-AP), anti-Akt (Cell Signaling Technology, Boston, MA, USA, 88800SF), and, anti-Phospho-Akt (Cell Signaling Technology, Boston, MA, USA, 4060S).

### 4.5. Immunofluorescence Staining

The cells were seeded on climbing sheets on a 12-well plate. After culturing for 24 h, the cells were washed thrice with phosphate-buffered saline (PBS) and fixed in 4% paraformaldehyde for 20 min at room temperature. After washing thrice with PBS, the cells were incubated with 0.1% Triton X-100 at room temperature for 10 min. Blockade was achieved by incubating the cells with 5% bovine serum albumin (BSA) at room temperature for 60 min, followed by incubation with a 1:200 dilution of primary antibodies at 4 °C overnight. The samples were then washed thrice with PBS, followed by incubation with 1:1000-diluted Alexa Fluor 488(563)-conjugated secondary antibodies (yeasen, Shanghai, China) at room temperature for 60 min. Finally, the cell samples were stained with 4ʹ,6-diamidino-2-phenyl-indole (DAPI) (meilunbio, Dalian, China) for 15 min at room temperature. Images were captured using a DM2500 fluorescence microscope (Leica, Weztlar, Germany). The following primary antibodies were used: anti-MORTALIN (Cell signaling tech-nology, Boston, MA, USA, 3593S), anticleaved CASPASE3 (Cell signaling technology, Boston, MA, USA, 9664), anti-p53 (Proteintech, Chicago, IL, USA, 60283-2-Ig), and anti-p65 (Proteintech, Chicago, IL, USA, 10745-1-AP).

### 4.6. Wound Healing Assay

The ovarian cancer cells were seeded on a six-well plate. When the cells reached 90–100% confluency, a sterile 10 μL pipette tip was used to scratch the cells to form a straight line at the bottom of the 6-well plates. The cells were washed thrice with PBS to remove exfoliated cells and cultured in fresh DMEM. Wound closure was visualized using a DM2500 fluorescence microscope (Leica, Weztlar, Germany) at 0 and 48 h. The wound healing area was also calculated.

### 4.7. Transwell Assay

Cells (2 × 10^5^) were suspended in serum-free medium and seeded into the inserts of a 24-well Transwell chamber (Corning, NY, USA) in triplicate. The inserts were co-cultured on a 24-well plate containing 10% FBS as an attractant. After incubation at 37 °C in a 5% CO_2_ atmosphere for 24 h, the inserts were removed, and the upper surface of the membrane was scrubbed to remove the non-migrating cells. The inserts were then fixed with 4% paraformaldehyde and stained with crystal violet staining solution (meilunbio, Dalian, China). Images of migrated cells were acquired, and the average cell counts were calculated.

### 4.8. Determination of Cell Viability

To determine cell proliferation 24 h after transfection with miR-200b/c or NC, cells were digested and reseeded in a 96-well plate at a density of 1000 cells per well. The cells were then incubated with the CCK-8 solution, which was diluted in DMEM containing 10% FBS, for 1 h, and the absorbance at 450 nm was measured using a Multiskan MK3 microplate reader (Thermo Fisher Scientific, Waltham, MA, USA). For the drug resistance assay, 24 h after transfection with miR-200b/c or NC, cells were digested and reseeded on a 96-well plate at a density of 5000 cells per well and treated with successive concentrations of cisplatin (0, 10, 20, and 40 μM) for another 24 h. Cell viability was assessed by incubating the cells with CCK-8 solution.

### 4.9. Flow Cytometry Analysis

Apoptosis was detected via flow cytometry using the Annexin V-fluorescein isothiocyanate (FITC)/ PI apoptosis detection kit (meilunbio, Dalian, China). Briefly, the digested cells were washed twice with PBS and resuspended in 1× binding buffer. The cells were incubated with Annexin V-FITC and PI for 15 min in the dark. After washing with cold PBS, the cells were resuspended in 400 μL of 1× binding buffer and subjected to fluorescence-activated cell sorting analysis using a flow cytometer (BD Biosciences, Franklin Lakes, NJ, USA).

### 4.10. Xenograft Tumor Growth Study

All animal experiments were approved by the Animal Experimentation Ethics Committee of Fudan University, Shanghai, China (No. 20170223-079). A2780CP (5 × 10^6^) cells in 50 μL PBS were mixed with 50 μL Matrigel (CORNING, NY, USA) and injected into the armpits of nude mice to develop a subcutaneous ovarian cancer model. The length and width of the transplanted tumors in nude mice were measured every three days using a digital caliper. Mice that did not successfully develop subcutaneous tumors were excluded, and the mice were divided into four groups: (1) no treatment; (2) intraperitoneal injection of cisplatin and intratumoral injection of normal saline (NC); (3) intraperitoneal injection of cisplatin and intratumoral injection of agomiR-200b; and (4) intraperitoneal injection of cisplatin and intratumoral injection of agomiR-200c. When the subcutaneous tumor volume reached 100 mm^3^, cisplatin was intraperitoneally injected into the abdominal cavity of nude mice every three days at a dose of 3 mg/kg. Three days after the first cisplatin treatment, agomiR-200b/c was administered to the tumor lump every three days at a dose of 3 nmol/mouse. After 22 days, the mice were sacrificed, and the tumors were removed. The tumor tissues were weighed and used for subsequent experiments.

### 4.11. Immunohistochemistry

Paraffin-embedded ovarian tumor tissue sections were immersed twice in dimethylbenzene for 10 min for deparaffinization. Paraffin sections were then immersed in ethanol (100%, 95%, 85%, and 75%) for 5 min for hydration. Antigen retrieval was achieved by boiling sections in an antigen retrieval solution for 5 min. Endogenous peroxidase activity was quenched with 3% hydrogen peroxide diluted in methanol for 15 min in the dark. The sections were subsequently incubated with 1% BSA for 20 min at room temperature to block the antigen. The sections were incubated with primary antibodies (1:100 in 1% BSA) overnight at 4 °C, followed by incubation with horseradish peroxidase-labeled goat anti-rabbit IgG for 1 h at room temperature in a wet box. DAB chromogenic solution (Dako, Copenhagen, Denmark) was added to the tissues, and the reaction was terminated using ddH2O at the appropriate time. Subsequently, the tissues were dehydrated in successive ethanol washes and the dimethylbenzene was used to remove the remaining ethanol.

### 4.12. Analysis of Clinical Ovarian Cancer Data from TCGA Dataset

Normalized mRNA and microRNA expression datasets for ovarian cancer [50] were downloaded from the cBioPortal for cancer genomics (http://www.cbioportal.org, accessed on 22 August 2022). This dataset included the mRNA and miRNA profiles of 489 ovarian cancer samples. The correlation between the overall survival (OS)/clinical stage and miR-200b/c transcript levels was analyzed. Pearson’s correlation coefficients were calculated for these transcripts in all ovarian cancer samples. *p* < 0.05 was considered to be statistically significant.

### 4.13. ChIP Assay

The binding of NF-κB p65 to the promoter region of *miR-200b/c* was detected using a ChIP assay kit (Beyotime, Shanghai, China), according to the manufacturer’s protocol. Briefly, the protein-DNA complex was immunoprecipitated overnight at 4 °C with the NF-κB p65 antibody (Cell Signaling Technology, Boston, MA, USA, 8242, 1:100). Samples immunoprecipitated with IgGs were used as negative controls. RT-qPCR was performed to detect the binding of NF-κB to the *miR-200b/c* promoter. All RT-qPCR primers are listed in the Appendix A. *miR-200c* and *PTPN6* share the same promoter, so the sequence of the *miR-200c* promoter is the 5′ end of *PTPN6*.

### 4.14. Statistical Analysis

Statistical analyses were performed using GraphPad Prism 8.0 software, and a graphical representation of the data has been illustrated. Data are presented as mean ± standard deviation (SD) from at least two independent experiments. Student’s t-tests and two-way analysis of variance (ANOVA) were used to determine significant differences. Sidak’s test with two-way ANOVA was used for multiple comparisons. Differences were considered statistically significant at *p* < 0.05. The correlation coefficient between the two parameters was evaluated using Pearson’s correlation test. The log-rank test was used to detect differences in clinical prognosis.

## Figures and Tables

**Figure 1 ijms-23-15300-f001:**
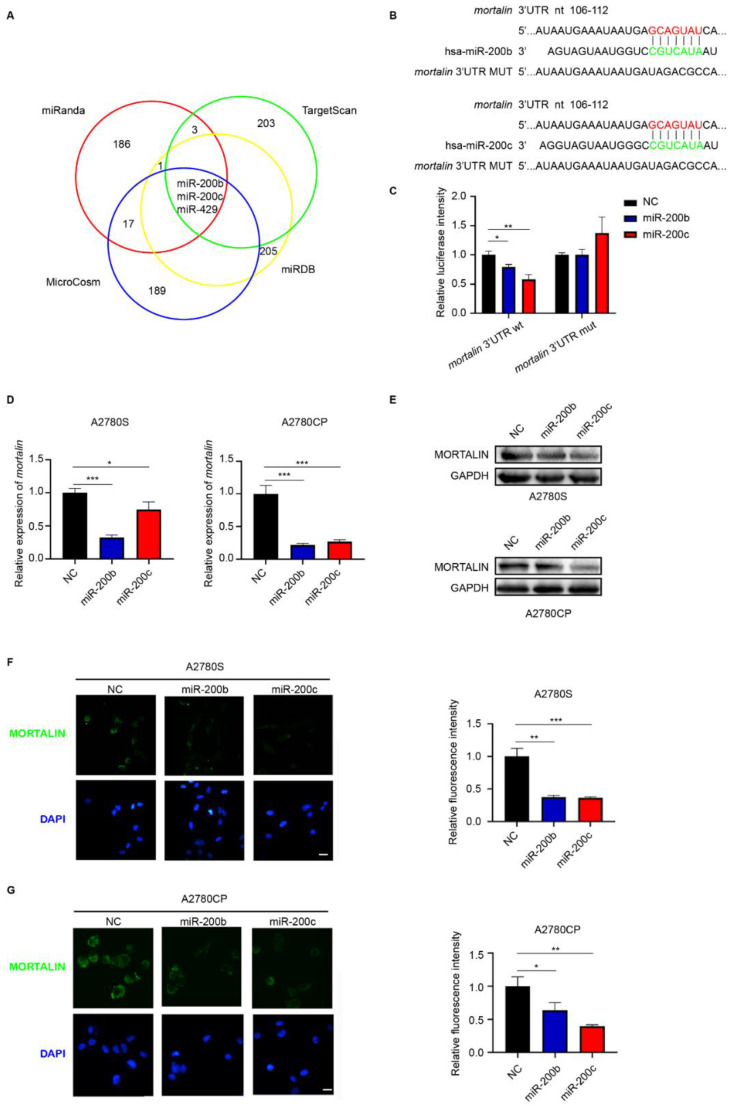
Mortalin is a direct target of microRNA (miR)-200b/c. (**A**) Venn chart depicts the intersection of the miRNAs predicted to regulate mortalin expression by four different miRNA databases; four circles represent the four different databases: TargetScan (green), miRanda (red), miRDB (yellow), and MicroCosm (blue). (**B**) The potential binding site of miR-200b/c on the 3′-UTR region of *mortalin* coding sequence and mutated sequence of the binding site. (**C**) Luciferase reporter gene assay detected the relative fluorescence intensity of firefly luciferin in 293T cells after co-transfection of miR-200b/c mimic or negative control sequence (NC) with pmiRGLO *mortalin* 3′-UTR or pmiRGLO *mortalin* 3′-UTR mutated recombinant plasmids. (**D**) mRNA expression of *mortalin* was determined via quantitative polymerase chain reaction (qPCR) after transfection of A2780S and A2780CP cells with miR-200b/c mimic or NC. (**E**) Protein expression of mortalin was determined via Western blotting after transfection of A2780S and A2780CP cells with miR-200b/c mimic or NC. (**F**) Expression and localization of mortalin were analyzed using immunofluorescence after transfection of A2780S cells with miR-200b/c mimic or NC (scale bar, 10 μm). (**G**) Expression and localization of mortalin were analyzed using immunofluorescence after transfection of A2780CP cells with miR-200b/c mimic or NC (scale bar, 10 μm). Biological replicates (n = 3). * *p* < 0.05, ** *p* < 0.01, *** *p* < 0.001.

**Figure 2 ijms-23-15300-f002:**
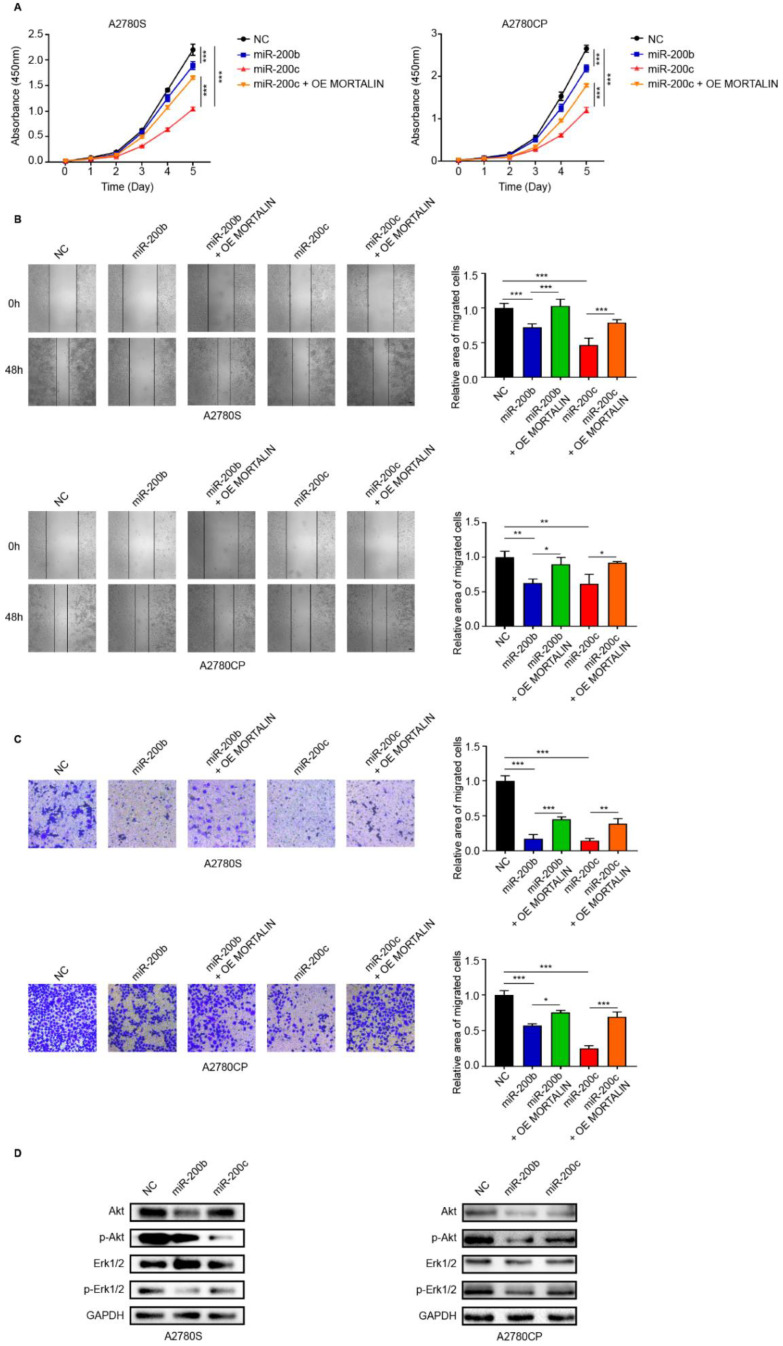
miR-200b/c decreases proliferation and migration of ovarian cancer cells via mortalin. After transfection of A2780S and A2780CP cells with miR-200b/c mimic or NC, (**A**) cell viability was determined using the CCK-8 assay. (**B**) Representative images from the wound healing assay of different groups at 0 and 48 h (scale bar, 50 μm). Quantification of the wound-healing area. (**C**) Migration of ovarian cancer cells was determined via transwell assay (scale bar, 20 μm). Average cell counts of 10 randomly selected fields. (**D**) Protein levels of Erk1/2, p-Erk1/2, Akt, and p-Akt were determined via Western blotting. Biological replication (n = 3). * *p* < 0.05, ** *p* < 0.01, *** *p* < 0.001.

**Figure 3 ijms-23-15300-f003:**
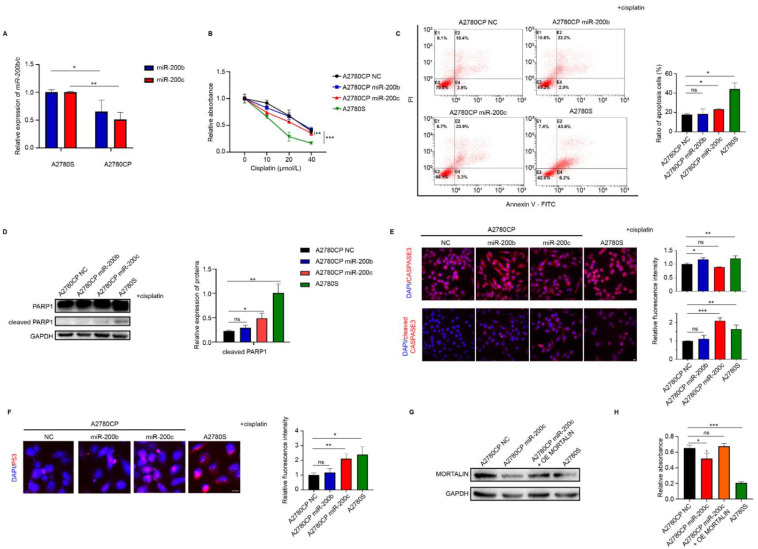
miR-200c increases cisplatin sensitivity of ovarian cancer cells. (**A**) Transcript levels of *miR-200b/c* in A2780S and A2780CP cells were determined via qPCR. Ovarian cancer cells were treated with different concentrations of cisplatin after transfection of A2780CP cells with miR-200b/c mimic or NC for 24 h. (**B**) Cell viability was determined via CCK-8 assay. (**C**) The levels of apoptosis markers, annexin V and propidium iodide (PI), were determined by measuring the mean fluorescence intensity using flow cytometry. (**D**) Expression levels of apoptosis-related proteins, PARP-1, and cleaved PARP-1, were determined via Western blotting. (**E**) Immunofluorescence assay to determine the expression of caspase-3 and cleaved caspase-3 (red fluorescence), and 4′,6-diamidino-2-phenylindole (DAPI; blue fluorescence) to determine the location of nucleus (scale bar, 10 μm). (**F**) Immunofluorescence assay to determine the expression of p53 (red fluorescence) and its translocation into the nucleus, and DAPI to determine its location in the nucleus (scale bar, 10 μm). (**G**) Western blotting showing the protein expression levels of mortalin in A2780CP cells after transfection of miR-200c mimic alone or co-transfection with miR-200c mimic and mortalin overexpression plasmid. (**H**) After transfection for 24 h, cells were treated with 20 μM cisplatin for 24 h. Additionally, cell viability was determined via CCK-8 assay. Biological replicates (n = 3). * *p* < 0.05, ** *p* < 0.01, *** *p* < 0.001, ns *p* > 0.05.

**Figure 4 ijms-23-15300-f004:**
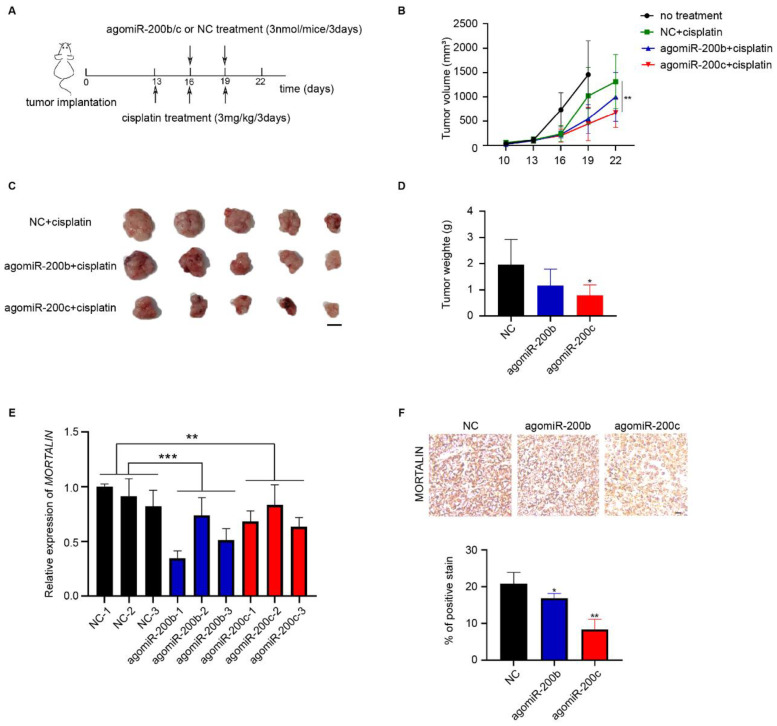
miR-200c suppresses chemoresistance of ovarian cancer cells in vivo. (**A**) Schematic diagram of the experimental design of a subcutaneous tumorigenesis model in nude mice. (**B**) After subcutaneous tumor inoculation for 10 d, the subcutaneous tumor volume was measured every three days to plot the tumor growth curve. (**C**) Images of isolated tumors on day 22 (scale bar, 1 cm). (**D**) Weights of the isolated tumors. (**E**) mRNA expression levels of *mortalin* in ovarian cancer tissues treated with agomiR-200b/c or NC were determined by qPCR. (**F**) The expression of mortalin in isolated tumor tissues were determined using immunohistochemistry (scale bar, 10 μm). Biological replicates (n = 5). * *p* < 0.05, ** *p* < 0.01, *** *p* < 0.001.

**Figure 5 ijms-23-15300-f005:**
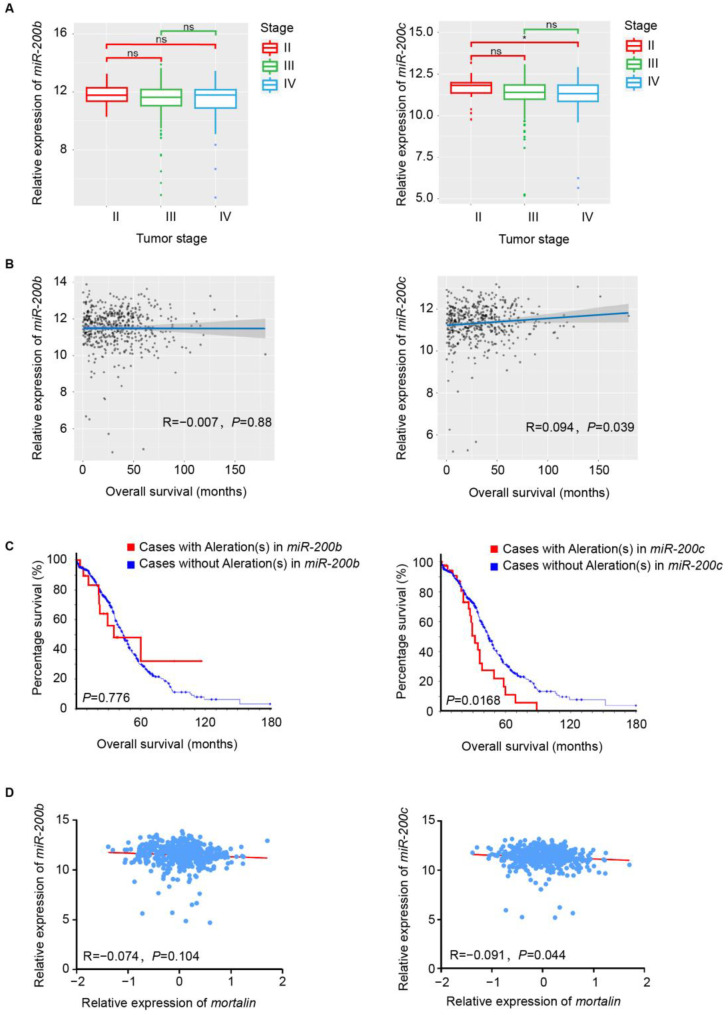
*miR-200c* expression levels are associated with patient prognosis. (**A**) Box diagram shows the correlation between *miR-200b/c* expression levels and different clinical stages (stages II, III, and IV) in TCGA 2011 ovarian cancer data (n = 489). (**B**) Scatter plot shows the correlation between *miR-200b/c* expression levels and the overall survival of patients with ovarian cancer. (**C**) Kaplan–Meier analysis indicates the correlation between *miR-200b/c* gene mutation and the overall survival of patients with ovarian cancer. (**D**) Scatter plot shows the correlation between *miR-200b/c* and *mortalin* expression levels in patients with ovarian cancer. * *p* < 0.05, ns *p* > 0.05.

**Figure 6 ijms-23-15300-f006:**
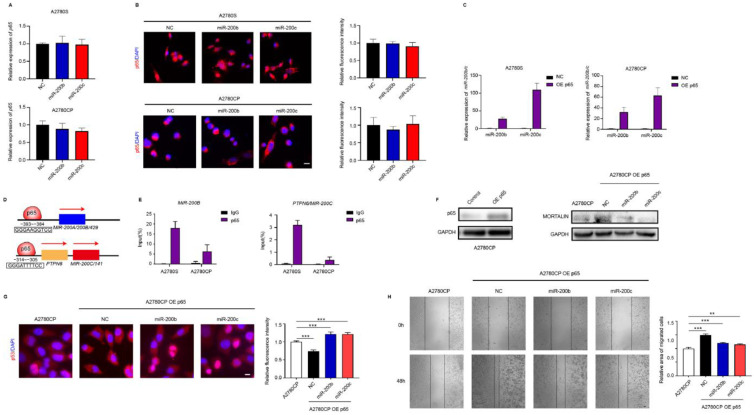
NF-κB and miR-200b/c co-regulate mortalin expression. (**A**) mRNA expression levels of *mortalin* were determined via qPCR after transfection of A2780S and A2780CP cells with the miR-200b/c mimic or NC. (**B**) Nuclear translocation of NF-κB was detected using immunofluorescence after miR-200b/c transfection (scale bar, 10 μm). (**C**) Expression levels of *miR-200b/c* were determined via qPCR after transfection of ovarian cancer cells with the NF-κB overexpression plasmid. (**D**) Predicted NF-κB binding site in the promoter region of *miR-200b/c*. (**E**) ChIP assay was used to detect the binding of NF-κB to the promoter of *miR-200b/c*. (**F**) miR-200b/c mimic or NC was transfected into A2780CP NF-κB stable overexpression cell line, and the protein expression levels of mortalin were determined via Western blotting. (**G**) Nuclear translocation of p53 was detected using immunofluorescence (scale bar, 10 μm). (**H**) Migration of ovarian cancer cells was determined via transwell assay (scale bar, 50 μm). Average cell counts of 10 randomly selected fields. Biological replicates (n = 3). ** *p* < 0.01, *** *p* < 0.001.

**Figure 7 ijms-23-15300-f007:**
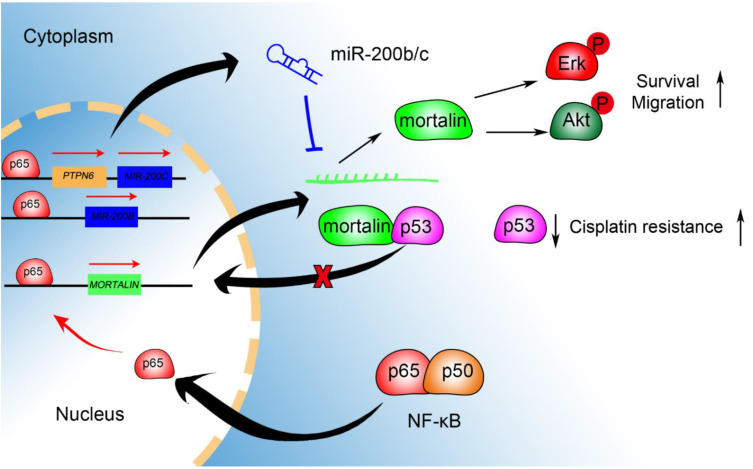
Summary model. Schematic representation of the regulation of mortalin expression and its downstream bi0logical effects.

## Data Availability

The data underlying this article will be shared on reasonable request to the corresponding author.

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
