# Peer review of "A Regulatory Loop Involving miR-200c and NF-κB Modulates Mortalin Expression and Increases Cisplatin Sensitivity in an Ovarian Cancer Cell Line Model"

_ijms, 2022, doi:10.3390/ijms232315300_

Round 1
Reviewer 1 Report
This manuscript investigates the regulation of Mortalin expression by miR200b/c and NF-kB in A2780 ovarian cancer cells and a chemotherapy resistant derivative of this same line. They show that mortalin expression is decreased in both cell lines using miR200b, and more so with miR200c, and this leads to decreased viability, wound healing, migration, sensitization to cisplatin. The topic is interesting and important, however the manuscript suffers from several weaknesses, most notably the use of a single cell line that has a chemosensitive version and a chemoresistant derivative. This is a serious deficiency given the heterogeneity that characterizes ovarian cancer. There are also some controls and rescue experiments missing. Many of the experiments rely on overexpression studies which are not complemented by knockdown experiments. The regulatory loop cannot be established without additional functional data that verifies the mechanism.
Abstract:
1) It does not seem correct to say that ovarian cancer is common. It is not among the top 10 for incidence in women. Perhaps focusing on its prognosis is better. For example, it is the 5th leading cause of cancer related deaths in women.
Methods:
1) The use of armpit as a subcutaneous injection site is unusual. Can the authors clarify why a traditional flank injection was not used?
2) The use of control mice that do not receive any drug are important positive controls for tumor growth yet they do not appear to be included in the animal studies.
3) A dose curve using a wide range of concentrations to generate IC50 values rather than a few successive doses of cisplatin would be more informative for calculating chemoresistance.
4) The use of one cell line and its derivative is not rigorous. Unfortunately, many of the conclusions drawn from this study cannot be supported by the current data. It is important to distinguish biological effects that are reproducible across cell lines and this manuscript only evaluates the A2780 sensitive and chemoresistant line which may produce cell-line specific behavior that has no relevance to ovarian cancer, which is a notoriously heterogenous disease.
Results:
1) The rationale for looking into miRNAs in the regulation of Mortalin is somewhat unclear. The authors have previously shown that NF-kB regulates Mortalin expression at the promoter but then states that miRNAs can also contribute to gene regulation. Did the previous studies suggest additional regulatory factors are required for Mortalin expression?
2) It’s unclear why the luciferase reporter was used in the 293T cells instead of the ovarian cancer cells
3) Figure 2C-D would benefit from rescue experiments to complement the data presented in A-B.
4) Line 285, cisplatin is not likely the most used chemotherapy drug as carboplatin and paclitaxel is the standard of care. Please provide a reference for this claim.
5) Supplementary Figure 3C should be in the primary figures, not the supplement.
6) Figure 3: It's unclear why the chemosensitive line was not included in these studies. There is Mortalin expression in the A2780S cells and this could help differentiate the role of miR200b/c and Mortalin expression in chemoresistance
7) Figure 3F: The text implies chemoresistance, however the figure legend only implies viability without any indication of the specific conditions. Overall, the data is not convincing and should include a time course, a dose curve, and a calculation of IC50
8) Figure 4: These experiments should have vehicle control groups? It's unclear whether the agomirR would function similarly in the absence of cisplatin
9) Figure 6: The overexpression vector should be verified that NF-kB is positively expressed (western blot). E-F should demonstrate NF-kB overexpression. Also, this figure would be significantly strengthened if some of the functional assays from early figures (migration, p53 translocation, etc) were done in this context to confirm mechanistic features.
Discussion:
1) It's unclear from this study which is more important for regulating Mortalin - miR-200b/c or NF-kB?
Author Response
We thank the reviewer for correctly understanding our work and considering our topic interesting and important. And we have supplemented several data and revised the manuscript extensively based on the reviewer’s suggestion. We also sought the assistance of MDPI English editing, a professional manuscript editing service, for further editing and grammar corrections. We hope that the revised manuscript is significantly improved. We thank the reviewer once again for the constructive comments. The detailed corrections are listed below.

Reviewer 2 Report
Xin Huang, et al., reported that the miR-200c and NF-kB regulates mortalin expression and increases cisplatin sensitivity in ovarian cancer. Authors explore the role of miR-200b/c in down-regulation of mortalin expression and inhibit the proliferation and migration of ovarian cancer cells. Finally, they concluded the combination of both cisplatin and miR-200c significantly enhanced the therapeutic effect on ovarian cancer in nude mice. I think authors are well elaborated the results and discussion part.
Here is some of the minor comments to make the publication clearer and better understanding.
1. In methods section line 85, what are the 293T, is it ovarian cancer cells?
2. In fig, 4 C, Why the tumor size is various within the same group (NC + Cis)?
Author Response
We thank the reviewer for correctly understanding our work and considering our work well elaborating on the results and discussion part. It is encouraging for us.

Reviewer 3 Report
Huang et al present data concerning the roles of p65 and miR-200 family miRNAs in regulating mortalin and platinum response in ovarian cancer. This is a clear expansion of this group’s previous work in this field; the addition of miRNA to the network is the novel aspect of this manuscript. The paper is overall clearly written, although some editing for English is needed. The study is well structured and the authors include both in vitro and in vivo studies in support of their claims. However, a graph appears to be duplicated, and extensive edits and clarifications are needed prior to publication. These are outlined below, along with other suggestions the authors may consider:
Materials and Methods:
-Line 85: what is meant by the cell lines being “saved in our laboratory?” Please provide source if purchased or gifted, or other authentication if not.
-Line 94: was the pmirGLO reporter co-transfected with the miR mimic? Or stably integrated in the cells? Pease clarify order and timing of transfections.
-Lines 120, 133, 135: Please provide catalog numbers for antibodies for western and IF studies.
-Line 125: authors state that westerns were quantified using ImageJ, but not quantitation is given. Either quantification should be added or this note should be removed.
-Line 184: the design of the animal protocol is concerning. There is no non-treated or miRNA only (no cisplatin) control. This may be justified if the objective is simply to show miRNA treatment sensitizes the tumors to cisplatin, thus cisplatin only serves as the control. However, if this is the case, why is the miRNA treatment started AFTER cisplatin treatment has already begun? Authors may also wish to justify the use of a subcutaneous model for ovarian cancer.
-Line 201: why is there a second deparaffinization step listed?
-Line 214: please provide p65 antibody catalog number and dilution.
-Line 223: please clarify if tests for multiple comparisons were performed as part of statistical analyses.
-Primers: There are multiple issues with the primers provided in the supplementary document. The main text states that beta actin is used as a housekeeping control for mRNA and U6 for miRNA targets. The supplementary table shows GAPDH in place of actin and no U6 primers at all. The mortalin primers as given are in the same orientation and the reverse primer comes before the forward in the sequence. The p65 forward primer does not match p65 according to BLAST, and the reverse only has 15/24 identities. No primer sequences are given for qPCR of miR200b or miR200c, only for their promoters for ChIP. The miR200b promoter reverse primer only has 21/24 identities according to BLAST. Finally, the miR200c promoter primers show correct orientation but appear to be 5’ of PTPN6. Figure 6D suggests both transcripts are produced from one promoter, so this would be correct, but please clarify in the text.
Results:
-Line 232: miR4499 is shown in Supp Fig 1 but never discussed in the text.
-Figure 1F,G: please provide length of scale bar in the figure legend as done for other images.
-In all figure legends, authors may wish to specify statistical analyses done in addition to p-values.
-Line 265: Mortalin overexpression only partially rescues proliferation – please clarify.
-Figure 2B: The two graphs shown in this panel appear to be identical, including the error bars, despite having different statistical results shown. Please replace whichever graph is in error with the correct data.
-Line 269: Figure 2C is not mentioned in the text.
-Line 274: total Akt also appears to be reduced, not just the phospho-Akt. Why was no data shown from A2780S? Please explain or include these data.
-Figures 3A-C show quite small changes. Without quantitation, the western in 3C is particularly hard to draw conclusions from. In addition, the authors should discuss the large loss of total caspase 3 following MiR200b/c expression.
-Line 300: authors may with to mention that miR200b does not affect p53 either, referring the reader to Figs 3B and 3D.
-Line 321: the authors introduce agomirs with no further explanation. A reference here would be helpful.
-Figure 4E: it is not clear what conditions are being compared for statistics. The lines over the groups should be edited to make this clear to the reader.
-Line 340: miR-200c is lower in stage IV than II, but not III, at least according to the statistics. Please clarify.
-Line 346: similarly, only miR200c shows statistical significance in Fig 5D. Please clarify.
-Figure 6A: the authors may wish to use “p65” rather than “NFKB” as the y-axis label.
Discussion:
-Line 402: the authors here misstate that miR-200c is higher in late stage tumors, it should be lower.
-Line 421-425: the authors discuss the differences between 200b and 200c, but an additional fact should be acknowledged: Although miR-200b shows less downregulation of mortalin protein in Fig 1E, Supp Fig 1 shows more downregulation, and the rest of Fig 1 supports this. Therefore, both miRNAs downregulate mortalin, but only miR-200c affects apoptosis and 200c has a much greater effect on proliferation and in vivo tumor growth. Therefore, it may be that some of this effect of miR-200c is independent of mortalin. The authors should discuss this. Also, since only miR-200c affects p53, p53 may be responsible for the changes in growth.
-Figure 7: the p53 interaction in the figure should be discussed in the main text, possibly around Line 429 where p53 is talked about. Conversely, the authors may wish to add the effect on cisplatin response discussed in the main text to the figure.
-A final point: Figure 7 appears to show direct binding between mortalin and p53. Authors may wish to include a CoIP study to prove this, or reference their prior study if this has already been done.
Author Response
We thank the reviewer for correctly understanding our work and considering our paper overall clearly written. And we also sought the assistance of MDPI English editing, a professional manuscript editing service, for further editing and grammar corrections. We sincerely apologize for duplicated graph in the main text, the graph in error has been replaced in the revised manuscript. We thank the reviewer once again for the constructive comments. We believe that the revised manuscript is significantly improved.

Round 2
Reviewer 1 Report
Unfortunately the revisions are not comprehensive enough to support the conclusions drawn. There are still controls missing and entire study relies on the use of one cell line.
Author Response
We thank the reviewer for this important comment. To make the article more convincing, we have made the following changes. First, we have added the no treatment control group into Figure 4B. Then, we have changed our title to: A regulatory loop involving miR-200c and NF-κB modulates mortalin expression and increases cisplatin sensitivity in an ovarian cancer cell line model (Line 2). Finally, we have also explained the study limitation in the Discussion section and recalled it in the Abstract in the revised manuscript (Line 21 and Line 599). We thank the reviewer again for helping us to increase the readability of our manuscript.
Reviewer 3 Report
Huang et al have substantially improved their manuscript following incorporation of comments from multiple reviewers. I have only two remaining suggestions for the preparation of the final version:
1) The authors state that for their in vivo experiment, non-treated control animals all had to be euthanized on day 19, hence the exclusion of data on these animals. I agree that comparison of groups at the 22 day time point is appropriate for much of the data shown in Figure 4. However, for the graph in Figure 4B, including the control group up to day 19 would be a valuable comparison.
2) The authors have expanded their discussion to include many important analyses. However, the section describing miRNA and transcription factor networks, especially lines 566-570, remains difficult to understand. Some minor text edits to clarify would be helpful.
Author Response
We thank the reviewer for these important comments. The detailed corrections are listed in the attachment.
